# Physics-Informed Deep Learning for Fast Cerebral Perfusion Assessment in Stroke Digital Twins

Sebastiaan Broos[*1], Thijs Kuipers[1,2], and Henk Marquering[1,2]

[1]Amsterdam UMC, Department of Biomedical Engineering and Physics
[2]Amsterdam UMC, Department of Radiology and Nuclear Medicine
s.c.a.broos@amsterdamumc.nl

## 1 Abstract

In acute ischemic stroke, reduced blood flow decreases cerebral perfusion. This lowers tissue oxygenation and triggers cell death that can result in severe neurological impairment. Finite element methods (FEM) are used to study cerebral perfusion and simulate blood flow patterns in stroke, but their long computation times limit clinical use. We propose a physics-informed neural surrogate based on a sinusoidal representation network (SIREN) that reproduces Laplacian perfusion physics. The SIREN uses a lightweight encoder that embeds mesh geometry and estimates the normalized harmonic distance from the ventricles to the cortex and the corresponding blood flow directions. In held-out patients, it achieved FEM-level accuracy (MAE 0.034 on a 0–1 scale, 9.5° mean angular deviation) while achieving a 500 times reduction in computation time from 19 s to 0.036 s.

## 1 Introduction

Every minute is critical in acute ischemic stroke care, where treatment delays can have irreversible consequences for patient outcomes [1]. Digital twins have emerged as a promising tool to support clinical decision-making by enabling patient-specific simulation of treatment strategies. The Horizon Europe GEMINI consortium [2] develops multi-scale digital twins for ischemic and hemorrhagic stroke by integrating data-driven and mechanistic models, including finite element method (FEM)-based perfusion modeling [3, 4]. These twins simulate stroke pathophysiology, treatments, and outcomes. A key component is perfusion modeling, which estimates tissue blood distribution to identify at-risk regions and predict treatment outcomes. FEM solvers compute these perfusion fields by solving partial differential equations (PDEs), but their runtime limits clinical applicability. In the GEMINI stroke digital twin, the full FEM simulation is computationally intensive, with the perfusion model alone taking about five minutes [3]. To overcome this computational bottleneck, we propose a physics-informed neural

surrogate based on sinusoidal representation networks (SIREN) [5]. The surrogate replaces the first PDE of the FEM perfusion model. This PDE computes the normalized harmonic distance from the ventricles to the cortex ($p$) and its negative gradient $\mathbf{e}_{loc}$ to define the local blood flow directions that guide downstream perfusion and transport computations.

## 2 Methods

**Data.** The SIREN model, which serves as a surrogate for the first PDE of the FEM perfusion model, was trained on subject-specific tetrahedral meshes of the brain derived from MRI imaging of 75 healthy older adults in the EPAD cohort [4, 6]. Gray matter, white matter, and ventricles were segmented in T1–weighted MRI images. A tetrahedral brain template from the IXI555 Montreal Neurological Institute atlas [7] was affine-registered to each subject using FLIRT [8] to obtain a subject-specific mesh. Each mesh contains 299,585 vertices and 1,427,274 tetrahedra. Data were split patient-wise into 60/10/5 train/validation/test sets.

**Baseline FEM Perfusion Solver.** The FEM perfusion solver numerically models blood and oxygen transport through brain tissue by solving seven PDEs for perfusion and flow [3, 4]. We targeted the first PDE that models arterial inflow and venous outflow through porous brain tissue. In this stage, the solver computes the normalized harmonic distance field $p$ by solving the Laplace equation $\nabla^2 p = 0$ on the patient mesh with Dirichlet conditions $p=1$ on the cortex and $p=0$ on the ventricles. This field represents the normalized distance from the ventricles and provides a smooth gradient that defines the local blood flow directions as $\mathbf{e}_{loc} = -\nabla p/\|\nabla p\|$. We quantified intrinsic accuracy limits (error floors) of the FEM solution. For $p$, a residual-jump estimate was used, and for the flow direction $\mathbf{e}_{loc}$, we evaluated one-level mesh refinement and recomputation.

**Models.** Our pipeline consists of a geometric encoder module and a physics-informed SIREN. It estimates $p$ and $\mathbf{e}_{loc}$ from a patient-specific brain mesh in two stages. In the first stage, the encoder captures local mesh structure and global spatial

---
[*]Corresponding author: s.c.a.broos@amsterdamumc.nl

context. In the second stage, the SIREN estimates $p$ from this geometric representation.

*Geometry encoding.* We computed 16 per-vertex features that describe spatial location, local shape, and mesh context. An encoder [9] learned a 60D latent from supervision on $(p, \mathbf{e}_{\mathrm{loc}})$.

*SIREN surrogate.* A physics-informed SIREN [5] estimated $\hat{p}$ from per-vertex inputs: normalized coordinates $(x, y, z)$, boundary one-hot labels, the 16 features, and the 60D latent. Training minimized a weighted sum of four losses: data loss w.r.t. FEM-derived $p$, Dirichlet penalties for cortex = 1 and ventricles = 0, a Laplacian residual loss $\Delta\hat{p} \rightarrow 0$, and a cosine loss aligning $-\nabla\hat{p}$ with FEM $\mathbf{e}_{\mathrm{loc}}$. Directional alignment was prioritized due to downstream reliance on $\mathbf{e}_{\mathrm{loc}}$.

*Training setup & Evaluation.* We trained the model in PyTorch using mixed precision on a single V100 GPU. Hyperparameters (width 512, depth 9, frequency $\omega_0=2$, and learning rate $10^{-3}$) were chosen based on validation performance. We evaluated the SIREN surrogate on held-out patients ($N=5$) and report normalized-distance error, flow-direction alignment, and trajectory consistency with the FEM reference, which assesses whether blood flow paths follow the same routes from the cortex to the ventricles. Trajectory consistency was measured by tracing 2000 paths per patient along the estimated $\mathbf{e}_{\mathrm{loc}}$ field and comparing their endpoints and path lengths to FEM-derived trajectories. We also report inference time relative to the FEM baseline.

# 3 Results & Discussion

**Results.** Error floors were found at 0.0135 on a 0–1 scale for $p$ and at 9.1° for $\mathbf{e}_{\mathrm{loc}}$. Table 1 reports error metrics for $p$ and $\mathbf{e}_{\mathrm{loc}}$. Figure 1(a) shows the error distribution of $p$ and Figure 1(b) shows the angular deviation of $\mathbf{e}_{\mathrm{loc}}$. Table 1 also reports blood flow trajectory metrics with respect to the FEM-solver. The SIREN reduced the harmonic normalized-distance computation time from 19.36 s to 0.036 s ($\times533$) and the end-to-end computation to $\mathbf{e}_{\mathrm{loc}}$ from 66.74 s to 35.29 s ($\times1.89$).

**Discussion.** The physics-informed surrogate accurately reproduced the FEM field $p$ while respecting Laplacian and Dirichlet constraints. The $p$ errors were slightly above the FEM $p$-error floor, yet remained relatively small. Errors were concentrated in regions where the distance between the ventricles and cortical surface was smallest, such as in the posterior occipital horns, where steep gradients in $p$ are expected. In these areas, $\mathbf{e}_{\mathrm{loc}}$ remained closely aligned with the FEM reference, as training prioritized correct gradient orientation over exact $p$ values. Larger angular deviations occurred near the septum pellucidum, since multiple attractors make the direction ambiguous. As this region contributes little to

**Table 1.** Held-out test results ($N=5$). Metrics cover field accuracy and flow-trajectory agreement.

| Metric | Value |
|---|---|
| *Field accuracy* | |
| $p$ (MAE / $R^2$) | 0.034 / 0.964 |
| $\mathbf{e}_{\mathrm{loc}}$ (angle / cosine) | 9.5° / 0.979 |
| Boundary MAE (cortex / ventricles) | 0.026 / 0.003 |
| Physics residual (mean $(\Delta\hat{p})^2$) | $\sim 10^{-12}$ |
| *Flow trajectory agreement with FEM* | |
| Endpoint $\Delta$ (hops: med / mean) | 1 / 2.39 |
| Overlapping path proportion (len frac) | 0.53 |
| $\Delta$ number of steps (med / mean) | 0 / 0.46 |

**(a)**

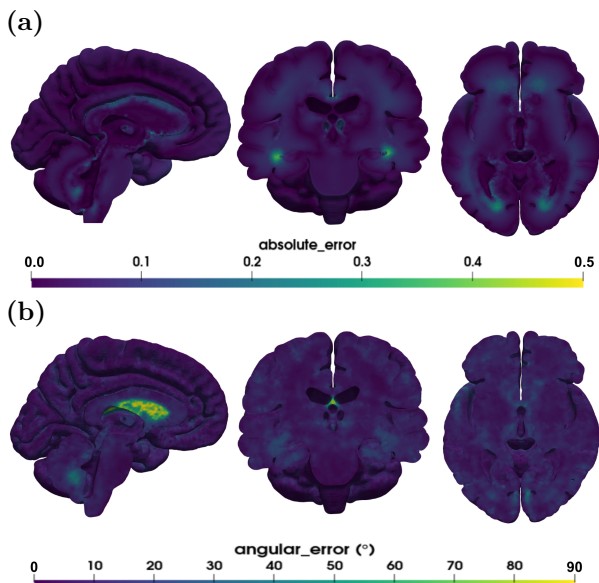

**(b)**

**Figure 1.** Error analyses. (a) $p$ error maps on representative axial, sagittal, and coronal slices. (b) Angular error for $\mathbf{e}_{\mathrm{loc}}$ with respect to the FEM direction field.

perfusion estimation, the effect on downstream computations is negligible. Trajectory analysis indicated that global transport behavior was preserved. Endpoints, path lengths, and routes were consistent with the FEM reference. Inspecting the runtimes, most time was spent on geometric feature preprocessing. This step is performed once per patient mesh and reused across modules, limiting its overall impact and indicating potential for major speedups once the entire FEM perfusion solver is surrogated. Future work will therefore focus on developing surrogates for the six remaining PDEs in the FEM perfusion solver to achieve a fully accelerated pipeline fit for clinical applications. In conclusion, the surrogate reproduced the FEM distance and flow estimates with high fidelity while achieving significant runtime reductions. It preserved physiologically consistent flow patterns and marks a key step toward a fully surrogate-based GEMINI pipeline enabling real-time what-if treatment exploration for clinical decision support.

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
