# OpenReview forum: "Physics-Informed Deep Learning for Fast Cerebral Perfusion Assessment in Stroke Digital Twins"
_NLDL.org/2026/Abstracts_Track — NLDL 2026 Abstracts_

### Official Review · Reviewer_LN8Q · 2025-10-28

**Soundness:** 4
**Correctness:** 4
**Rating:** 5
**Confidence:** 4

**Summary:**

The authors propose using a physics-informed neural surrogate based on a sinusoidal representation network that reproduces Laplacian perfusion physics to mitigate the high computational costs associated with finite element methods in studying cerebral perfusion and simulating blood flow patterns in stroke.

**Strengths:**

The abstract is well written and results are shown on a very challenging real life dataset.
Data-driven approaches to mitigate the computational costs associated with physics modeling are highly relevant for many applications, hence this contribution will be interesting to a broader audience beyond the application in the focus of this study.

**Weaknesses:**

The authors do not mention that code for the method is being released which would be very valuable in terms of reproducibility and future research in physics-informed ML.

---

### Official Review · Reviewer_fvvg · 2025-11-03

**Soundness:** 3
**Correctness:** 3
**Rating:** 4
**Confidence:** 4

**Summary:**

This paper presents a physics-informed deep learning (PIDL) approach for accelerating cerebral perfusion modeling in digital twins for ischemic stroke. Traditional finite element method (FEM) solvers accurately simulate blood flow and oxygen transport through brain tissue but are computationally expensive, limiting real-time clinical use.

To overcome this, the authors design a sinusoidal representation network (SIREN)-based surrogate that replaces the first PDE in the FEM perfusion model — specifically the Laplace equation for normalized harmonic distance from ventricles to cortex (p) and its gradient-derived flow direction (eloc).

The model combines a geometric encoder (extracting spatial and mesh-context features) with a physics-informed SIREN that enforces Dirichlet boundary conditions and Laplacian residual constraints while aligning estimated gradients with FEM-based flow directions.

Trained on MRI-derived tetrahedral meshes from 75 subjects, the proposed surrogate achieves near-FEM accuracy (MAE = 0.034, angular deviation = 9.5°) and an impressive 500× speedup (19 s → 0.036 s). Qualitative and trajectory analyses demonstrate that the surrogate preserves global flow consistency and physiological plausibility. The authors note potential for further acceleration by extending this surrogate framework to all PDEs in the FEM solver, moving toward a fully real-time, physics-informed digital twin for stroke treatment simulation.

**Strengths:**

Addresses a major bottleneck in stroke digital twin modeling of the excessive runtime of FEM-based perfusion simulations.

The surrogate is not a black-box network; it explicitly encodes Laplacian physics, Dirichlet boundary conditions, and flow direction constraints, ensuring physically consistent outputs. This is a well-executed example of physics-informed deep learning (PIDL).

The use of sinusoidal representation networks to model spatially continuous, differentiable perfusion fields is both elegant and effective for mesh-based physiological modeling.

Extensive metrics (MAE, angular deviation, flow trajectory agreement) and visualizations (error maps, angular deviation maps) convincingly show that the surrogate preserves critical flow characteristics.

The 500× reduction in runtime while maintaining near-FEM accuracy is a remarkable engineering and methodological achievement, demonstrating the feasibility of real-time digital twin updates.

The paper clearly details data preprocessing, model architecture, loss formulation, and evaluation metrics, ensuring reproducibility and interpretability.

**Weaknesses:**

- The model was validated on only five held-out patients from a healthy aging cohort, not on stroke patients, where altered anatomy and flow conditions may challenge model robustness. This raises concerns about clinical generalizability.

- The proposed method replaces only one of seven PDEs in the FEM solver. While an important step, the end-to-end acceleration remains limited (1.89× overall). The paper’s conclusions about clinical readiness are somewhat premature until the full pipeline is surrogated.

- A notable portion of runtime remains in mesh feature computation, which somewhat offsets the claimed real-time performance and could complicate practical deployment.

- The study does not compare against other surrogate modeling methods (e.g., graph neural networks, PINNs, or U-Net-based FEM surrogates). Such comparisons would better contextualize the advantages of SIREN for this task.

---

### Decision · Program_Chairs · 2025-11-05

**Decision:**

Accept

**Comment:**

The abstract is of interest to the community and should be presented at the conference.